# *Striga hermonthica* Suicidal Germination Activity of Potent Strigolactone Analogs: Evaluation from Laboratory Bioassays to Field Trials

**DOI:** 10.3390/plants11081045

**Published:** 2022-04-12

**Authors:** Muhammad Jamil, Jian You Wang, Djibril Yonli, Tsuyoshi Ota, Lamis Berqdar, Hamidou Traore, Ouedraogo Margueritte, Binne Zwanenburg, Tadao Asami, Salim Al-Babili

**Affiliations:** 1The BioActives Lab, Center for Desert Agriculture, King Abdullah University of Science and Technology (KAUST), Thuwal 23955-6900, Saudi Arabia; muhammad.jamil@kaust.edu.sa (M.J.); jianyou.wang@kaust.edu.sa (J.Y.W.); lamis.berqdar@kaust.edu.sa (L.B.); 2Institut de l’Environnement et de Recherches Agricoles (INERA), Ouagadougou 04 BP 8645, Burkina Faso; d.yonli313@gmail.com (D.Y.); hamitraore8@yahoo.com (H.T.); margoued616@gmail.com (O.M.); 3Applied Biological Chemistry, The University of Tokyo, 1-1-1 Yayoi, Bunkyo, Tokyo 113-8657, Japan; higenomoto.co@gmail.com (T.O.); asami@mail.ecc.u-tokyo.ac.jp (T.A.); 4Institute for Molecules and Materials, Radboud University, 6525 AJ Nijmegen, The Netherlands; b.zwanenburg@science.ru.nl; 5Plant Science Program, Biological and Environmental Science and Engineering Division, King Abdullah University of Science and Technology (KAUST), Thuwal 23955-6900, Saudi Arabia

**Keywords:** germination stimulant, witchweed, methyl phenlactonoates (MPs), Nijmegen-1, weed

## Abstract

The obligate hemiparasite *Striga hermonthica* is one of the major global biotic threats to agriculture in sub-Saharan Africa, causing severe yield losses of cereals. The germination of *Striga* seeds relies on host-released signaling molecules, mainly strigolactones (SLs). This dependency opens up the possibility of deploying SL analogs as “suicidal germination agents” to reduce the accumulated seed bank of *Striga* in infested soils. Although several synthetic SL analogs have been developed for this purpose, the utility of these compounds in realizing the suicidal germination strategy for combating *Striga* is still largely unknown. Here, we evaluated the efficacy of three potent SL analogs (MP3, MP16, and Nijmegen-1) under laboratory, greenhouse, and farmer’s field conditions. All investigated analogs showed around a 50% *Striga* germination rate, equivalent to a 50% reduction in infestation, which was comparable to the standard SL analog GR24. Importantly, MP16 had the maximum reduction of *Striga* emergence (97%) in the greenhouse experiment, while Nijmegen-1 appeared to be a promising candidate under field conditions, with a 43% and 60% reduction of *Striga* emergence in pearl millet and sorghum fields, respectively. These findings confirm that the selected SL analogs appear to make promising candidates as simple suicidal agents both under laboratory and real African field conditions, which may support us to improve suicidal germination technology to deplete the *Striga* seed bank in African agriculture.

## 1. Introduction

Cereal crops, particularly maize (*Zea mays*), sorghum (*Sorghum bicolor*), and pearl millet (*Pennisetum glaucum*), are important nutrient and livelihood sources for people in the region of sub-Saharan Africa [1]. However, the average yield of these cereals in Africa is very low, as compared to the world average. Increasing the per hectare yield of crops is one of the major challenges to alleviate hunger and poverty in sub-Saharan Africa [2]. Indeed, the agriculture practiced by the majority of smallholder farmers in Africa is facing a wide range of threats, such as low soil fertility, recurrent drought, and many biotic constraints, particularly parasitic weeds. One of the most troublesome and noxious weeds is *Striga hermonthica* (*Striga*), which hinders crop production and causes more than 50% cereal yield losses in sub-Saharan Africa [3,4].

*Striga hermonthica* (also known as purple witchweed) is an obligate and hemiparasitic plant that parasitizes the root systems of cereals. It is now even becoming a persistent threat to crop production in sub-Saharan Africa, the Middle East, and parts of Asia [5,6,7,8]. It is estimated that *Striga* causes more than 50% cereal yield losses in sub-Saharan Africa and has already infested up to 40% (approximately 50 million hectares) of cropland in this region [9,10,11], thus building a large reservoir for further propagation and spreading of this weed. Although *Striga* has infested several regions of Africa, the most severely infested countries are Burkina Faso, Gambia, Mali, Niger, Nigeria, Senegal, Togo (West Africa), Ethiopia, Eritrea, Kenya, Mozambique, Malawi, Somalia, Tanzania (East Africa), and Sudan. Beyond those, *Striga* infestation is gradually spreading to other African countries, such as Cameron, Rwanda, Congo, Burundi, Zambia, and Zimbabwe. The ever-increasing *Striga* problem incurs an annual loss of US $7–10 billion per annum to the livelihoods of African farmers, and its elevated levels in vast areas will further worsen situations of food insecurity, hunger, and poverty for millions of subsistence farmers [12,13,14,15]. Hence, developing suitable *Striga* control strategies to minimize the damage is of urgent need for African agriculture.

Despite the debilitating yield losses caused by *Striga*, effective control measures are still far beyond success. Production of more than 200,000 tiny (0.3 nm  ×  0.15 nm), light (4–7 μg), and long-lived (>15 years) viable seeds by a single *Striga* plant, characterized by a complex life cycle and underground parasitism, makes *Striga* control burdensome [16,17]. Even though local farmers are aware of the *Striga* problem, many factors, such as poverty, limited resources, and harsh weather have impeded the control of this parasite in infested areas. In such cases, the key principles for effective control of *Striga* should be prevention of new seed production, significant reduction in the soil seed bank, and avoiding spreading of seeds from infested to non-infested soils [18,19,20,21].

The tiny *Striga* seeds have developed a host detection strategy by perceiving host-released signaling molecules, mainly the phytohormone strigolactones (SLs), which trigger their germination [22,23,24]. Strigolactones (SLs) are a group of evolutionarily conserved carotenoid-derived plant hormones that regulate many aspects of plant biology [25,26,27]. The well-known reason why plants release SLs is to successfully communicate with the surrounding arbuscular mycorrhiza (AM) for symbiotic interactions [23,28]. In the parasitic plant *Striga*, the receptors responsible for SL-induced seed germination are *Striga hermonthica* hyposensitive to light (ShHTLs), a group of α/β s that are assumed to be evolutionarily derived from a karrikin-insensitive2 (KAI2) receptor and to arise through gene duplication with functional replacement by SLs [29]. Among the several SL receptors, ShHTL7 is the most sensitive to SLs, suggesting a major role in modulating the seed germination of *Striga* [30,31]. Recently, ShHTL7 has been further characterized as a non-canonical receptor by noting how covalently linked intermediate molecule (CLIM), the SL hydrolysis intermediate, interacts with the F-box proteins ShMAX2 and SMAX1 to mediate SL signaling transduction [32], in a similar way as to rice D14. After successful germination and attachment to the host roots, *Striga* seedlings start siphoning off nutrients and water [33,34]. This dependency of *Striga* germination on host-released SLs and host attachment can be utilized as an effective management strategy [15,35]. The induction of *Striga* seeds’ germination through synthetic SL analogs in the absence of the host leads to the death of germinating seedlings at the latest a few days after germination [36,37]. This mechanism builds the basis for a promising control strategy, known as “suicidal germination”, which can reduce the *Striga* seed bank in the infested soils [38,39,40]. However, the assessment of utility and the employment of suicidal germination relies on the development of cheap and potent SL analogs. Although this technology has been suggested and advocated in several studies to combat not only *Striga* [37,38,39,40,41,42,43] but also other root parasites of the *Orobanchaceae* family, in the past 40 years, its application in real African fields remains contested. Despite the availability of several SL analogs and mimics, the selection of simple SL analogs to be used as potential *Striga* suicidal agents, especially under field conditions, is still a challenge. In the present study, we selected two potent methyl phenlactonoates (MP3 and MP16) [24,37] and a well-known, easy-to-synthesize SL analog, Nijmegen-1 [43], for a comparative study on inducing *Striga* seed germination, first under laboratory conditions.

## 2. Results

### 2.1. Striga Seed Germination Bioassays in Response to Various Strigolactone Analogs

The structures of the selected SL analogs are shown in Figure 1. All selected SL analogs at the 1 μM concentration showed a 49–52% *Striga* germination rate, which is statistically equal to the standard SL analog GR24 (64%) (Figure 2). However, *Striga* seeds were more sensitive to a lower concentration of GR24 (0.1 μM) with a maximum germination rate (66%), compared to the two other analogs at the same concentration (34–37%). Interestingly, higher concentrations, (1–10 μM) did not show any significant differences between SL analogs. Next, we applied these SL analogs to *Striga* seeds collected from Kenya, Burkina, and Niger to account for differences between *Striga* populations from different origins (Appendix A). At a concentration of 1 μM, GR24 and MP3 showed a higher germination rate (20–22%) than MP16 and Nijmegen-1 (14%) in the *Striga* seeds collected from Burkina Faso. In contrast, the four SL analogs exhibited statistically equal *Striga* germinating activity (11–16%) on the seeds collected from Kenya and Niger. These results unraveled differences in the sensitivity of *Striga* populations toward SL analogs but also indicated that the selected analogs could be candidates for testing and realizing the introduction of suicidal germination technology in various parts of Africa.

### 2.2. Striga Emergence on Strigolactone Analogs’ Treatment under Greenhouse Conditions

Surprisingly, all three SL analogs showed a significant reduction in *Striga* emergence in comparison to the control treatment (Figure 3A–C). The reducing effect of MP16 on *Striga* emergence was strongest (97%), followed by Nijmegen-1 (89%) and MP3 (73%). As expected, the application of SLs led to improved growth of the host plant due to lower infection of viable *Striga* plants. In particular, MP16-treated pots had the same plant heights as non-infested plants. Application of MP3 and Nijmegen-1 also improved plant growth as compared to the control treatment, but to a lesser extent than MP16, which indicated that the latter is the most potent suicidal agent among the selected analogs under greenhouse conditions (Figure 3C).

### 2.3. Effect of Strigolactone Analogs on Striga Infection under Field Conditions

In the pearl millet field, application of Nijmegen-1 led a smaller number of *Striga* plants to emerge (32 ± 6.1), followed by MP16 (33 ± 9.8) and MP3 (37 ± 6.7), as compared to the control plot (56 ± 16.4). In other words, treatment with Nijmegen-1, MP16, and MP3 led to a 43%, 41%, and 33% reduction of *Striga* emergence, respectively (Figure 4C). In the sorghum field, formulated Nijmegen-1 also exhibited the highest reduction of *Striga* emergence (60%), followed by formulated MP3 (52%) and MP16 (11%) (Figure 5C). With respect to the host plants, we observed a 38% increase in pearl millet grain yield and 64% for the stalk yield on MP16 application, whereas MP3 showed a moderate effect on grain yield (9%) and biomass yield (30%). Surprisingly, the application of Nijmegen-1 showed a reduction of pearl millet grain yield (13% less than blank), accompanied by a 31% increase in biomass yield (Figure 4D) In contrast, the application of this SL analog to sorghum led to the maximum observed increase in grain yield (13% higher than blank), while MP3 and MP16 did not show a pronounced effect (−4% and 3%, respectively). All the treatments showed a 1–37% increase in sorghum biomass yield over the blank treatment (Figure 5D). These results indicate the applicability of the selected SL analogs under real field conditions and their potential to solve the severe food security threats caused by *Striga*.

## 3. Discussion

The performance of three selected synthetic SL analogs (Figure 1) for *Striga* control was first assessed and confirmed using in vitro bioassays. The results showed that MP3, MP16, and Nijmegen-1 are all potent SL analogs inducing *Striga* seed germination (Figure 2). Previously, the moderate activity of Nijmegen-1 was reported for *Striga* germination [43,44,45], but our bioassays showed its comparable *Striga* germination activity at a high dose (1–10 μM). Similarly, we recently reported that MP3 and MP16 exhibit high bioactivity in stimulating *Striga* seed germination [24,37], which is in agreement with the outcome of this work. In addition, these SL analogs were also tested on different *Striga* batches collected from Burkina Faso, Niger, and Kenya. Although all SL analogs (at 1 μM) induce *Striga* seed germination ranging from 12–20%, this activity level is not as high as we observed in a *Striga* batch collected from Sudan. This result demonstrates that the sensitivity toward a certain SL analog depends on the ecotype of the *Striga* population. In addition, it shows the need to have an array of SL analogs to realize the suicidal germination strategy in different parts of Africa. Nevertheless, the results obtained with different *Striga* populations indicate that the selected SL analogs can be used in different regions of Africa, although with different efficiency levels.

The bioactivity of Nijmegen-1 was previously tested under pot conditions at a lower concentration (0.5 μM) and reported to be relatively weak [39]. In the present pot study, we modified the testing protocol by increasing the concentration of SL analogs to 1 μM and the number of applications to two. The results obtained indicated that increasing the dose and frequency of SL analogs might be an effective way to improve their efficacy as suicidal agents against *Striga*. The maximum 97% reduction in *Striga* emergence on the application of MP16 showed its potential to be used as a suicidal agent (Figure 3). Although we reported a high *Striga* germination-inducing activity of MP1, which carries a 4-nitro substituent on the benzene ring of MP3 [24,39], its synthesis is complex and expensive due to the involvement of additional synthesis steps and introduction of a 4-nitro group. So, alternately, we redesigned and synthesized another SL analog MP16, which is comparatively simple and easy to synthesize at a low cost. The *Striga* germination-inducing activity of the proposed MP16 was comparable to MP1. Similarly, the selected SL analogs might also potentially regulate the plant root architecture and above-ground shoot development when applied to a field [46]. However, there is still a lack of experimental evidence to support our understanding of the mechanisms involved. The actual molecular basis for the SL analogs on crop plants and crosstalk with other phytohormones still needs to be further investigated.

In field trials, we observed that all selected analogs caused a reduction in *Striga* emergence for both pearl millet and sorghum fields; however, these differences were statistically non-significant (Figure 4 and Figure 5). This might be attributed to the great variation in *Striga* infestation level between each plot. Moreover, the *Striga* seed density in the soil, soil texture, structure, pH, farmer’s cultural practices, and the amount of rainfall to the field are all possible factors affecting the observed variations between two fields and their replications. Weak activity of MP3 and Nijmegen-1 in clay soil and high in sandy loam-mixture soil has already been reported [39]. In addition, the inconsistency in the performance of MP3 and Nijmegen-1 for farmer’s fields has been indicated in a previous study [39], showing reductions of 55% and 65% in *Striga* emergence in a pearl millet field (sandy soil) but no significant impact (12–20%) in a sorghum field (clay soil). In the present study, we increased the number of applications from two to four and observed 33–43% and 60–52% reductions in *Striga* emergence by MP3 and Nijmegen-1 in a pearl millet and sorghum field, respectively. Due to seed dormancy or pre-conditioning requirements, *Striga* seeds are not synchronous in their germination response to SL analogs; hence, we suggested at least four-time repeated applications of SL analogs for maximum eradication of the seed bank in the infested soil. Then, the compound should reach *Striga* seeds to induce their germination. The success of the suicidal germination depends on many factors that are not restricted to the frequency and concentration of the suicidal agents but also include the type of the soil, climate conditions, and amount and frequency of rainfall [39,47]. In addition, it is very important to develop a handy, cost-effective, and stable formulation of suicidal agents that must be easily accessible to African farmers [48].

In summary, our results demonstrate the potential of the three selected simple SL analogs in combating *Striga*. The selected SL analogs will allow the development of further analogs as well as testing and optimizing of formulations and application protocols that aim at depleting the *Striga* seed bank and increasing food security. Moreover, combining the suicidal germination strategy with resistant crop varieties and/or suitable crop rotation or intercropping will likely alleviate the *Striga* problem significantly in the infested regions of Africa.

## 4. Materials and Methods

### 4.1. Plant Materials and Chemicals

The SL analogs *rac*-GR24, MP3, and Nijmegen-1 were synthesized and provided by Prof. Binne Zawanenburg, Radboud University, The Netherlands. The SL analog MP16 was synthesized and provided by Prof. Tadao Asami, University of Tokyo, Japan. The procedure for the synthesis of SL analogs has already been described previously [24,37,43,49]. *Striga hermonthica* seeds were collected from an infested sorghum field during 2020 in Sudan (Prof. A. G. Babiker), from a maize field during 2018 in Kenya (Prof. Steven Runo, Kenyata University), from a pearl millet field during 2020 in Burkina Faso (Dr. Djibril Yonli, INERA), and a pearl millet field during 2019 in Niger (Dr. Mohammed Riyazaddin, ICRISAT). Seeds of rice IAC-165 were obtained from Africa Rice, Tanzania (Dr. Jonne Rodenburg). The emulsifier Atlas G-1086, a polyoxyethylene sorbitol hexaoleate, was obtained from CRODA, The Netherlands.

### 4.2. Striga Seed Germination Bioassays

The germination activity of the selected SL analogs on *Striga hermonthica* seeds was determined by using a previously described procedure [50,51]. For pre-conditioning, *Striga* seeds were surface sterilized with 50% commercial bleach for seven minutes, followed by six washing steps with MiliQ water in a laminar fume hood. Then, about 50–100 surface-sterilized *Striga* seeds were uniformly spread on a glass-fiber filter-paper disc (9 mm), and 12 discs with *Striga* seeds were put in a Petri dish on sterilized filter paper moistened with 3 mL MiliQ water. The sealed Petri dishes were incubated in the dark at 30 °C for 10 days. On the 11th day, the discs were dried under a laminar flow cabinet, and the SL analogs (50 μL) were applied to each disc with various concentration ranges (10^−5^ M to 10^−7^ M). Being very active, the synthetic SL analog GR24 was used as a standard positive control instead of natural SL, which is very scarce in plants, unstable, and difficult to extract. After application, the *Striga* seeds were induced to germinate in the dark for 24 h at 30 °C. The discs were scanned under a binocular microscope and germinated, and non-germinated seeds were counted by SeedQuant [52], with a germination rate (in %) then calculated.

### 4.3. Striga Emergence under Greenhouse Conditions

To further evaluate their bioactivity, we tested the three SL analogs in *Striga*-infested pots under greenhouse conditions, using a highly susceptible rice cv. IAC 165 as a host crop. The SL analogs were applied twice at 1 μM concentration to artificially infested preconditioned *Striga* pots. After 10 days of application, three one-week-old rice seedlings were planted in each pot, and *Striga* emergence was counted at 10 weeks after sowing (WAS). A mixture of soil (Stender, Basissubstrat) with sand (3:1 ratio) was prepared [53] and about 0.5 L of this mixture without *Striga* seeds was added to the bottom of a 3-L perforated plastic pot. Then, about 20 mg (~8000) *Striga* seeds were thoroughly mixed in a 1.5-L soil mixture and added on the top of clean soil in the same pot. The *Striga* seeds were preconditioned in a greenhouse under hot (30 °C) and moist conditions for 10 days. On the 11th day, each pot was supplied with 500 ml (1 μM) SL analogs, and *Striga* seeds were allowed to germinate for suicidal death without a host for another 10 days. Then, three one-week-old rice seedlings (IAC-165, a *Striga*-sensitive variety) were planted in the middle of each pot. The rice plants were allowed to grow under normal growth conditions (30 °C, 65% RH, normal sunlight). After 10 weeks, *Striga* emergence was observed in each pot and compared with the mock treatment.

### 4.4. Striga Emergence under Field Conditions

Next, the suicidal germinating bioactivity of three selected SL analogs was further investigated under naturally infested rainfed farmer’s field conditions in Burkina Faso (Figure 4 and Figure 5). Formulated SL analogs were applied at a 1 μM final concentration four times on the onset of rainfall. The three selected candidate SL analogs were further evaluated under field conditions in eastern Burkina Faso. Two highly *Striga*-infested pearl millet and sorghum fields, located near the Kouaré research station (11°58′49″ N, 0°18′30″ E) of INERA (Institut de l’Environnement et de Recherches Agricoles), were selected. Trials were established in each farmer’s field, following a randomized complete block design with four independent replications using 4 m × 4 m (or 16 m^2^) plots. Each plot consisted of five (5) ridges/rows, and the distance between ridge/row was 0.80 m; all plots were spaced with four (4) ridges/rows to avoid any SL contamination through run-off. The emulsifier Atlas G-1086 was used to formulate MP3, MP16, and Nijmegen-1. Each SL analog was applied (25 mL/m^2^ at 400 µM) in formulated form four times in each field after the onset of rainfall (≥10 mm) to make a final concentration of 1 µM. A blank treatment (Atlas-G only) was included as the control. The experimental crops pearl millet (local cultivar Idipiéni) and sorghum (local cultivar Itchoari) were sown at least one week after the last application. The rainfall was 391.5 mm recorded in 25 days and 617 mm in 27 days during the pearl millet and sorghum growth period, respectively. Plots were weeded twice (15 and 30 days after sowing (DAS)) with hand hoeing before *Striga* emergence. Then, weeds other than *Striga* plants were hand-pulled until crop harvest. The emerged *Striga* plants were counted at 80 DAS, corresponding to the period of maximum emergence of *Striga* plants in the plots.

### 4.5. Statistical Analysis

Collected data were analyzed using statistical software package R (version 3.2.2) and GraphPad Prism (version 9.1.1). One-way analysis of variance (ANOVA) with the least significant difference (LSD) multiple range test and unpaired *t*-test were used to analyze the effect of different SL analogs on *Striga* infestation.

## Figures and Tables

**Figure 1 plants-11-01045-f001:**
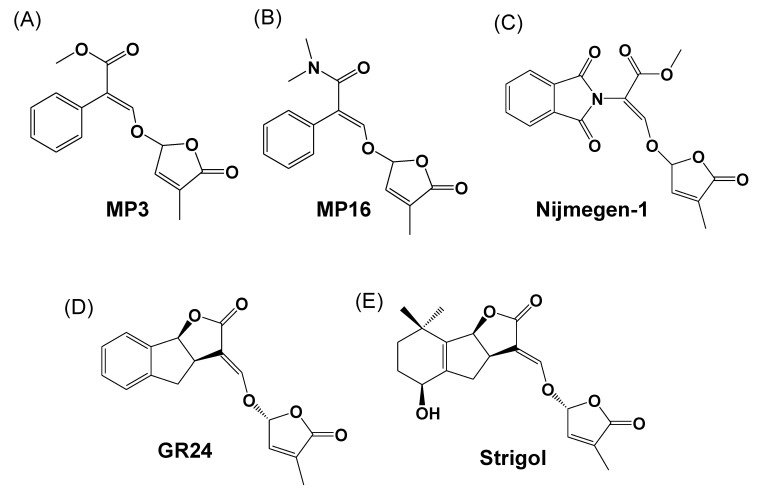
Chemical structures of strigolactone analogs (**A**) MP3, (**B**) MP16, (**C**) Nijmegen-1, (**D**) GR24, and (**E**) strigol, a natural strigolactone.

**Figure 2 plants-11-01045-f002:**
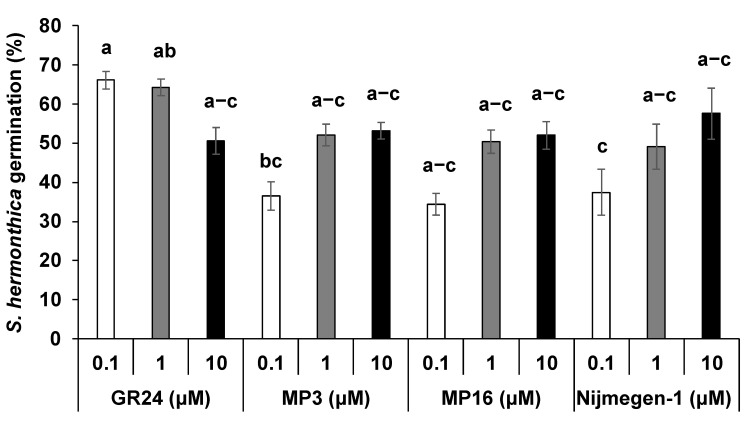
*Striga* seed germination rate in response to MP3, MP16, Nijmegen-1, and GR24. Various concentrations (10^−5^–10^−7^ M) of selected SL analogs were applied to 10-day preconditioned *Striga* seeds. GR24 was used as a positive control and water as a negative control. For each SL analog, treatments with various letters differed significantly according to one-way analysis of variance (ANOVA) and Tukey’s post-hoc test (*p* < 0.05). Error bars represent the standard error of the mean.

**Figure 3 plants-11-01045-f003:**
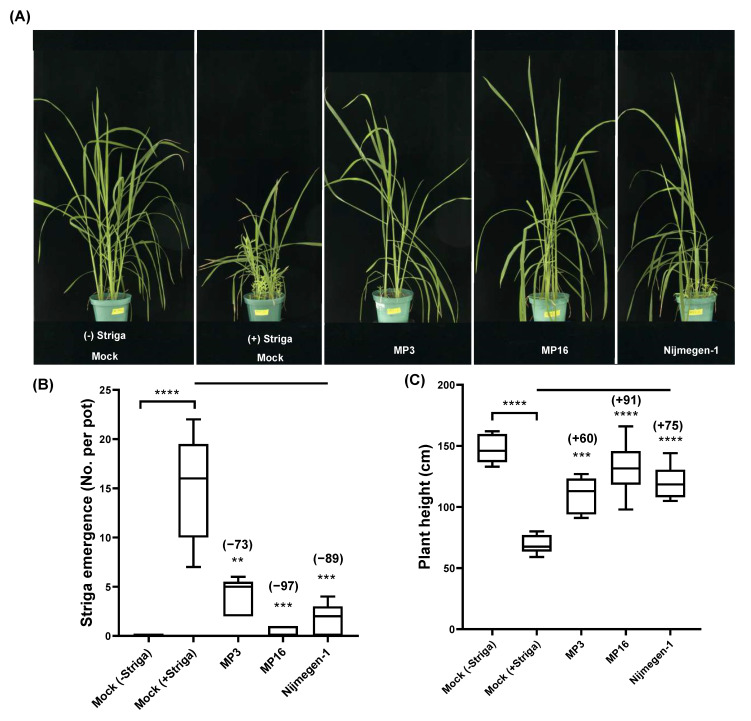
*Striga* emergence in response to the application of various SL analogs in pots under greenhouse conditions. (**A**) Representative pots of greenhouse study, showing *Striga* emergence under different treatments. (**B**) Number of *Striga* plants that emerged in mock, MP3, MP16, or Nijmegen-1 treatment. The selected SL analogs were applied twice at a 1 μM concentration to 10-day preconditioned *Striga*-infested pots. (**C**) Rice plant height in response to different treatments. Data are means ± SE (*n* = 6). Error bars represent the standard error of the mean. Values in parenthesis show the percentage increase (+) or decrease (−) over mock treatment. For each SL analog, treatments with various asterisks differ significantly according to one-way analysis of variance (ANOVA), ** *p* < 0.005, *** *p* < 0.005, **** *p* < 0.0005).

**Figure 4 plants-11-01045-f004:**
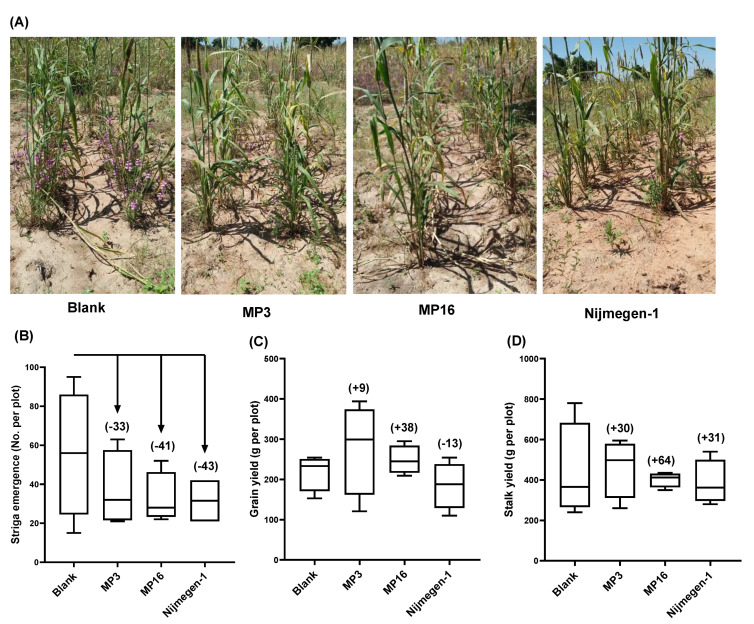
*Striga* emergence in response to different SL analogs for a naturally infested pearl millet field in Burkina Faso. (**A**) Representative pearl millet plots of farmer’s field, showing *Striga* emergence under different treatments. (**B**) Number of *Striga* plants that emerged in pearl millet field under mock, MP3, MP16, or Nijmegen-1 treatment. (**C**) Pearl millet grain yield per plot. (**D**) Pearl millet stalk yield per plot. Data are means ± SE (*n* = 4). Values in parenthesis show the percentage increase (+) or decrease (−) over mock treatment.

**Figure 5 plants-11-01045-f005:**
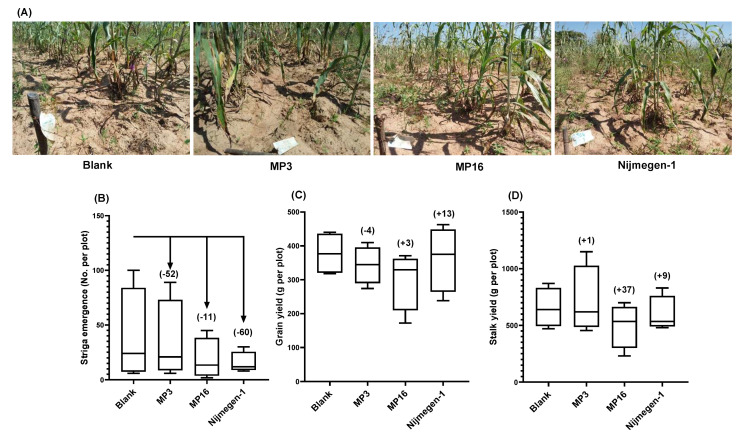
*Striga* emergence in response to different SL analogs for a naturally infested sorghum field in Burkina Faso. (**A**) Representative sorghum plots of farmer field, showing *Striga* emergence under different treatments. (**B**) Number of *Striga* plants that emerged in sorghum field under mock, MP3, MP16, or Nijmegen-1 treatment. (**C**) Sorghum grain yield per plot. (**D**) Sorghum stalk yield per plot. Data are means ± SE (*n* = 4). Values in parenthesis are showing the percentage increase (+) or decrease (−) over mock treatment.

## Data Availability

Not applicable.

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
