# Peer review of "Striga hermonthica Suicidal Germination Activity of Potent Strigolactone Analogs: Evaluation from Laboratory Bioassays to Field Trials"

_plants, 2022, doi:10.3390/plants11081045_

Round 1

Reviewer 1 Report

I agree that Striga hermonthica is the worst biological constraint to the staple food production in the sub-Saharan African countries. However, to my knowledge, this weed still stays within the African continent (Parker, C. Parasitic weeds: a world challenge. Weed Sci. 2012, 60, 269–276, doi:10.1614/ws-d-11-00068.1).

Due to its host dependent germination, induction of Striga seed germination in the absence of their hosts, induction of "suicidal germination", has been proposed as one of the promising strategies for its control. Unfortunately, efficacy of germination stimulants structurally related to natural stimulants strigolactones can be significantly influenced by environmental conditions, cultural practices, soil types and properties, amount and timing of rainfall, application methods and timings, etc. Therefore, as in the case of other herbicides and pesticides, field trials should be performed for at least two or three seasons to obtain reliable and reproducible data.

The authors already compared effectiveness of SL mimics MP1, MP3, and Nijmegen-1 in Striga suicidal germination in Petri dishes, pots, and rain-fed fields. In the present study, MP16 was used instead of MP1 but the results look very similar to those in the previous paper (Kountche et al. 2019). Therefore, the authors need to clarify what are important findings that are different from those in the previous paper.

Author Response

Response to Reviewer 1:

I agree that Striga hermonthica is the worst biological constraint to the staple food production in the sub-Saharan African countries. However, to my knowledge, this weed still stays within the African continent (Parker, C. Parasitic weeds: a world challenge. Weed Sci. 201260, 269–276, doi:10.1614/ws-d-11-00068.1).

Due to its host dependent germination, induction of Striga seed germination in the absence of their hosts, induction of "suicidal germination", has been proposed as one of the promising strategies for its control. Unfortunately, efficacy of germination stimulants structurally related to natural stimulants strigolactones can be significantly influenced by environmental conditions, cultural practices, soil types and properties, amount and timing of rainfall, application methods and timings, etc. Therefore, as in the case of other herbicides and pesticides, field trials should be performed for at least two or three seasons to obtain reliable and reproducible data.

The authors already compared effectiveness of SL mimics MP1, MP3, and Nijmegen-1 in Striga suicidal germination in Petri dishes, pots, and rain-fed fields. In the present study, MP16 was used instead of MP1 but the results look very similar to those in the previous paper (Kountche et al. 2019). Therefore, the authors need to clarify what are important findings that are different from those in the previous paper.

Response: We thank the reviewer for the valuable comments. We have responded and added information pertaining to this comment in the discussion.

Reviewer 2 Report

This is a very nice and interesting paper looking at alternative methods in Striga control which is well written and easy to follow.

However, I have some concerns on this manuscript. The weakest part of the manuscript is Discussion that need to be improved. I would start Discussion with main findings and explain them in relation to existing knowledge and emphasize novelty of the results from this study. However, there are only few references used for Discussion. Some sentences in Results fit much better to M&M. Those sentences are pointed below in minor corrections. It would be nice if the author can provide some ideas and future perspectives for Striga control based on the results they found in this study. There are some minor concerns that needs to be addressed. Conclusion part is long repeating what has already been mentioned without offering viewpoint based on the present state of this field of research.

Minor corrections:

Lines 16-17: I would rewrite this sentence as: The obligate hemiparasite Striga hermonthica is one of the major global biotic threats to agriculture in sub-Saharan Africa, causing severe yield losses of cereals.

Line 24: What do you mean by 50% Striga germination rate (reduction or increase)

Line 28: I am not sure what do you mean by

Line 30: Please check if you can use the same key words already mentioned in the title.

Line 50: “Although the distribution Striga covers…” Sentence is not clear. If I understood well you want to say that Striga spreads to several regions in Africa. Please consider rewriting this sentence.

Lines 94-95: Approach to what? Something is mission here. Please consider to rewrite whole sentence.

Lines 88-91: I would delete this sentence and move here sentence from the lines 95-98.

Lines 98-101: Those information are already mentioned in M&M. Here you should present results therefore I would delete those lines.

Line 110: “GR24 and MP3 showed a higher activity (20-22%)” What does it mean higher activity? Germination or something else.

Line 118. I would move Figure 1 to Supplementary material since you have not studied mode of action of these SL analogs

Line 120: I am not sure that scheme of the experiment is necessity to have. This is clearly explained in M&M. I would delete it. I would delete Figure 2c since you already have figure with results.

Line 122: In the line 98 the authors mentioned “water” and here H2O. Please uniform this through the whole text.

Lines 127-132: This belongs to M&M.

Line 143: In figure 3 I would delete experimental scheme. This is clearly explained in M&M. If Figure 3B and Figure 3C show the same results I would delete Figure 3B.

Lines 152-155: This should be part of M&M.

Line 158: Delete empty space before and after ±

Lines 174: Delete scheme of the experiment and Figure 4B.

Line 180: Delete scheme of the experiment and Figure 5B.

Lines 186-187: I think this sentence would better fit to Introduction.

Lines 187-188: I am not sure this sentence should be here, try to move it in Introduction or delete.

Lines 189-191: I would rather put this sentence at the end of Introduction or delete it because this sentence is repeated several times in the text.

Line 226: I am not sure you need to mention here reference 35.

Line 299: You do not have explanation for DAS abbreviation.

Line 304: Delete “statistically”.

Author Response

Reviewer 2:

This is a very nice and interesting paper looking at alternative methods in Striga control which is well written and easy to follow.

However, I have some concerns on this manuscript. The weakest part of the manuscript is Discussion that need to be improved. I would start Discussion with main findings and explain them in relation to existing knowledge and emphasize novelty of the results from this study. However, there are only few references used for Discussion. Some sentences in Results fit much better to M&M. Those sentences are pointed below in minor corrections. It would be nice if the author can provide some ideas and future perspectives for Striga control based on the results they found in this study. There are some minor concerns that needs to be addressed. Conclusion part is long repeating what has already been mentioned without offering viewpoint based on the present state of this field of research.

Response: We thank to the reviewer for these valuable comments. We have revised the manuscript and tried to address all the suggestions as suggested by the reviewer.

Minor corrections:

Lines 16-17: I would rewrite this sentence as: The obligate hemiparasite Striga hermonthica is one of the major global biotic threats to agriculture in sub-Saharan Africa, causing severe yield losses of cereals.

Response: Corrected as suggested.

Line 24: What do you mean by 50% Striga germination rate (reduction or increase)

Response: We have added the information.

Line 28: I am not sure what do you mean by

Response: We have rephrased the sentence.

Line 30: Please check if you can use the same key words already mentioned in the title.

Response: We have revised the keywords.

Line 50: “Although the distribution Striga covers…” Sentence is not clear. If I understood well you want to say that Striga spreads to several regions in Africa. Please consider rewriting this sentence.

Response: We have rewritten the sentence.

Lines 54-55: Approach to what? Something is mission here. Please consider to rewrite whole sentence.

Response: We have rewritten the sentence.

Lines 88-91: I would delete this sentence and move here sentence from the lines 95-98.

Response: We have revised the sentence.

Lines 98-101: Those information are already mentioned in M&M. Here you should present results therefore I would delete those lines.

Response: We have deleted the sentence.

Line 110: “GR24 and MP3 showed a higher activity (20-22%)” What does it mean higher activity? Germination or something else.

Response: We have revised the sentence.

Line 118. I would move Figure 1 to Supplementary material since you have not studied mode of action of these SL analogs

Response: We thank the reviewer for the suggestion. In our opinion, we should show the structure of SL analogs as main figure because we have compared mode of action (germination inducing activity) of these SL analogs.

Line 120: I am not sure that scheme of the experiment is necessity to have. This is clearly explained in M&M. I would delete it. I would delete Figure 2c since you already have figure with results.

Response: We have revised the figure.

Line 122: In the line 98 the authors mentioned “water” and here H2O. Please uniform this through the whole text.

Response: We have corrected as water.

Lines 127-132: This belongs to M&M.

Response: We have corrected as suggested.

Line 143: In figure 3 I would delete experimental scheme. This is clearly explained in M&M. If Figure 3B and Figure 3C show the same results I would delete Figure 3B.

Response: We have revised the figure.

Lines 152-155: This should be part of M&M.

Response: We have corrected as suggested.

Line 158: Delete empty space before and after ±

Response: We have deleted as suggested.

Lines 174: Delete scheme of the experiment and Figure 4B.

Response: Dear reviewer, we have deleted scheme of the experiment, but we still prefer to show picture of the field.

Line 180: Delete scheme of the experiment and Figure 5B.

Response: Dear reviewer, we have deleted scheme of the experiment, but we still prefer to show picture of the field.

Lines 186-187: I think this sentence would better fit to Introduction.

Response: We have revised as suggested.

Lines 187-188: I am not sure this sentence should be here, try to move it in Introduction or delete.

Response: We have revised as suggested.

Lines 189-191: I would rather put this sentence at the end of Introduction or delete it because this sentence is repeated several times in the text.

Response: We have revised as suggested.

Line 226: I am not sure you need to mention here reference 35.

Response: We have deleted the citation#35.

Line 299: You do not have explanation for DAS abbreviation.

Response: We have explained the abbreviation.

Line 304: Delete “statistically”.

Response: We have deleted as suggested.

Reviewer 3 Report

The authors describe the effect of several strigolactone analogs on germination activity of Striga hermonthica. Moreover, these physiological studies have been performed using laboratory, greenhouse and farmer's field conditions. This is an interesting study, but I have a few comments.

  1. Introduction section should contain some informations about molecular basis of strogolactones signaling (receptor or/and elements of signaling pathway).
  2. Results: I suggest that chemical structure of a major natural SL should be added to Fig.1
  3. GR24- a synthetic analog of natural SLs has been used as a positive control for experiments. Why ? And what about natural SL?
  4.  Discussion section. Molecular basis of the effect of SL on crop plants and cross-talk with other phytohormones should be discussed
  5. Material and methods: The preparation of syntethic analogs of SLs should be desribed.
  6. Further studies: This study focuses only on morphological parameters of plant. Application of SLs and interactions between hemiparasite and crop plant are associated with reactive oxygen species (ROS). I would to suggest that some ROS metabolites (e.g. ascorbate, glutathione, H2O2, lipid peroxides) and ROS signaling pathways (peroxidases, catalases, etc. ) could be analyzed during these studies.

Author Response

Reviewer 3:

The authors describe the effect of several strigolactone analogs on germination activity of Striga hermonthica. Moreover, these physiological studies have been performed using laboratory, greenhouse and farmer's field conditions. This is an interesting study, but I have a few comments.

Introduction section should contain some informations about molecular basis of strigolactones signaling (receptor or/and elements of signaling pathway).

Response: We have added in the introduction as suggested.

Results: I suggest that chemical structure of a major natural SL should be added to Fig.1

GR24- a synthetic analog of natural SLs has been used as a positive control for experiments. Why ? And what about natural SL?

Response: We have added the explanation.

Discussion section. Molecular basis of the effect of SL on crop plants and cross-talk with other phytohormones should be discussed

Response: We have added in the discussion.

Material and methods:

The preparation of synthetic analogs of SLs should be described.

Response: We have added information.

Further studies: This study focuses only on morphological parameters of plant. Application of SLs and interactions between hemiparasite and crop plant are associated with reactive oxygen species (ROS). I would to suggest that some ROS metabolites (e.g. ascorbate, glutathione, H2O2, lipid peroxides) and ROS signaling pathways (peroxidases, catalases, etc.) could be analyzed during these studies.

Response: We are thankful to the reviewer for this valuable comment. The suggested studies might need a lot of time and experimental set up and we will plan these studies in future to describe as another independent story.

Round 2

Reviewer 1 Report

The manuscript has been revised adequately.

There are some minor points need further revisions.

L 79; hermonthica should be in Italic

L 89; delete the sentence "After successful germination...." (duplicated)

L 95; the term "suicidal germination" was first coined by Elpee (Eplee, R.E. Ethylene: A witchweed seed germination stimulant. Weed Science 1975, 23, 433–436). It is better to refer to this.

L 102; potent methyl phenlactonoate (M should be m)

Figure 1. (E) Strigol should read strigol as it is a common name.

L 204, structure of MP1 should be explained; MP1 which carries 4-nitro substituent on the benzene ring of MP3.

L 206; and the introduction of 4-nitro group.

Author Response

Reviewer 1

Dear Reviewer

Thank you for reviewing our manuscript entitled “Striga hermonthica Suicidal Germination Activity of Potent Strigolactone Analogs: Evaluation from Laboratory Bioassays to Field Trials” by Jamil et al. for publication in the PLANTS.

We have addressed and incorporated all the suggestions raised.

L 79; hermonthica should be in Italic

Response: Corrected.

L 89; delete the sentence "After successful germination...." (duplicated)

Response:  Deleted.

L 95; the term "suicidal germination" was first coined by Elpee (Eplee, R.E. Ethylene: A witchweed seed germination stimulant. Weed Science 1975, 23, 433–436). It is better to refer to this.

Response:  Reference added.

L 102; potent methyl phenlactonoate (M should be m)

Response: corrected.

Figure 1. (E) Strigol should read strigol as it is a common name.

Response: corrected.

L 204, structure of MP1 should be explained; MP1 which carries 4-nitro substituent on the benzene ring of MP3.

Response: explained.

L 206; and the introduction of 4-nitro group.

Response: Corrected.